# Immunosuppression of Tumor-Derived Factors Modulated Neutrophils in Upper Tract Urothelial Carcinoma Through Upregulation of Arginase-1 via ApoA1-STAT3 Axis

**DOI:** 10.3390/cells14090660

**Published:** 2025-04-30

**Authors:** Chih-Chia Chang, Chia-Bin Chang, Cheng-Huang Shen, Ming-Yang Lee, Yeong-Chin Jou, Chun-Liang Tung, Wei-Hong Lai, Chi-Feng Hung, Meilin Wang, Ya-Yan Lai, Pi-Che Chen, Shu-Fen Wu

**Affiliations:** 1Department of Radiation Therapy and Oncology, Ditmanson Medical Foundation Chiayi Christian Hospital, Chiayi 60002, Taiwan; 07229@cych.org.tw; 2Department of Urology, Ditmanson Medical Foundation Chiayi Christian Hospital, Chiayi 60002, Taiwan; bining1029@hotmail.com (C.-B.C.); 01712@cych.org.tw (C.-H.S.); 07458@cych.org.tw (W.-H.L.); cfhung1017@gmail.com (C.-F.H.); 3Department of Hematology and Oncology, Ditmanson Medical Foundation Chiayi Christian Hospital, Chiayi 60002, Taiwan; 05825@cych.org.tw; 4Department of Urology, St. Martin De Porres Hospital, Chiayi 60069, Taiwan; 01729@cych.org.tw; 5Department of Pathology, Ditmanson Medical Foundation Chiayi Christian Hospital, Chiayi 60002, Taiwan; cych02112@gmail.com; 6Department of Microbiology and Immunology, School of Medicine, Chung Shan Medical University, Taichung 40201, Taiwan; wml@csmu.edu.tw; 7Department of Biochemical Science and Technology, National Chiayi University, Chiayi 60004, Taiwan; 01194@cych.org.tw; 8Department of Biomedical Sciences, and Epigenomics Human Disease Research Center, National Chung Cheng University, Chiayi 62102, Taiwan

**Keywords:** upper tract urothelial carcinoma, tumor tissue culture supernatant, neutrophils, arginase-1, T cells

## Abstract

Upper tract urothelial carcinoma (UTUC) presents aggressive features and a tumor microenvironment with T cell depletion. However, the role of tumor-associated neutrophils in UTUC remains unclear. This study aimed to investigate how UTUC tumor-derived factors modulate neutrophils and their impact on T cell immune responses. Our findings demonstrate that UTUC secreted tumor-derived factors, with apolipoprotein A1 (Apo-A1) being the predominant factor, which upregulated arginase-1 expression in neutrophils. STAT3 activation was responsible for the upregulation of arginase-1 in neutrophils. Blocking the interactions between Apo-A1 and its receptors reduced arginase-1 expression in neutrophils treated with tumor tissue culture supernatant (TTCS). Moreover, both CD4^+^ T and CD8^+^ T cell proliferation were inhibited by neutrophils treated with Apo-A1 or TTCS. Importantly, blocking Apo-A1 signaling in neutrophils reversed the inhibitory effects on T cells. In UTUC patients, the neutrophil-to-lymphocyte ratio was higher than that in healthy subjects. The expression of arginase-1 in neutrophils and the level of Apo-A1 within UTUC tumors were negatively correlated with tumor-infiltrating CD4^+^ T cells. Additionally, neutrophils from UTUC patients showed increased expression of arginase-1 and exhibited inhibitory effects of T cell functions. These findings suggest that UTUC orchestrates an immune-suppressive microenvironment through Apo-A1-mediated upregulation of arginase-1 in neutrophils, ultimately leading to the inhibition of T cell proliferation.

## 1. Introduction

Cancer treatment has made significant advancements in recent years, particularly through the revolutionized approach of cancer immunotherapy. PD-1/PD-L1-targeted immunotherapy has had a profound impact on cancer treatment, leading to long-lasting clinical benefits across various types of tumors [1]. Given that PD-1 serves as an inhibitory receptor for T cell response, tumor cells can upregulate PD-1 ligands to attenuate T cell-mediated anti-tumor immunity [2]. Despite the promise of PD-1/PD-L1-targeted therapies, only a subset of cancer patients experience significant clinical benefit from these treatments [3]. A contributing factor to this limited efficacy is the presence of immunosuppressive cells in the tumor microenvironment (TME), such as regulatory T cells, myeloid-derived suppressor cells (MDSCs), tumor-associated macrophages (TAMs), and neutrophils, all of which collectively foster an immune-suppressive environment that inhibits effective antitumor immunity [4]. Additionally, metabolic reprogramming of immune cells, driven by lipid signaling within the TME, further alters their function, promoting immune evasion and therapy resistance [5]. Specifically, tumor-associated immunosuppressive cells can render T cells ineffective and contribute to resistance against PD-1/PD-L1-based immunotherapies [6].

Upper tract urothelial carcinoma (UTUC) is a type of urothelial carcinoma that is known to exhibit more aggressive features compared to bladder cancer. Approximately 60% of UTUC cases are invasive at diagnosis, whereas bladder cancer represents only 15–25% with aggressive features [7]. The molecular characterization of UTUC predominantly displays luminal–papillary phenotype, which is associated with a T cell-depleted contexture in TME [8]. Understanding the cellular and molecular mechanisms that shape the TME in UTUC is therefore critical for developing more effective and targeted therapeutics for this cancer type.

The neutrophil-to-lymphocyte ratio (NLR) in peripheral blood is a prognosis marker in many cancers [9,10]. Higher preoperative and postoperative NLRs in UTUC patients are associated with poor clinical outcomes [11,12,13], suggesting a link between neutrophils and UTUC. Neutrophils are the most prevalent circulating immune cells competent for host defense against infection. Recently, increasing studies have illustrated the dual role of neutrophils in cancer progression [14]. Factors released from neutrophils, such as arginase-1, MMP-9 (matrix metalloproteinase-9), or VEGFs, can promote angiogenesis and tumor development. On the other hand, neutrophils also express TRAIL or ROS that can inhibit tumor development [15].

The peripheral immature neutrophils and granulocyte MDSCs (G-MDSCs) exert immunosuppressive properties, and their numbers tend to increase as cancer progresses [15]. Tumor-derived factors, including cytokines and lipids secreted by stromal cells, drive the recruitment and differentiation of these suppressive immune cells within the TME. These cells exhibit inhibitory effects on T cell activity and positively correlate with tumor progression and metastasis [16,17]. In bladder cancer, tumor-derived GM-CSF has been shown to activate neutrophils, inducing the expression of PD-L1 and inhibiting T cell activation [18]. Targeting infiltrated neutrophils in mouse bladder tumors has been explored as a potential strategy to enhance the efficacy of immune checkpoint inhibitors [19]. Similarly, GM-CSF derived from breast cancer could lead to T cell suppression through the induction of arginase-1 expression in myeloid cells [20]. Arginase-1 catalyzes the breakdown of arginine into ornithine and urea, leading to the metabolic changes that suppress T cell proliferation [21,22]. Moreover, the inhibition of arginase-1 production in neutrophils has been investigated as a therapeutic approach to induce hyperactivation of anergic T cells from multiple myeloma patients [23]. This evidence underscores the critical role of neutrophils, particularly through the expression of PD-L1 and arginase-1, as mediators of tumor-induced T cell suppression.

The use of metabolites derived from lipids, such as those from lipoprotein hydrolysis or fatty acid oxidation, is implicated in the immune suppression of macrophages and neutrophils [5]. For example, fatty acid transport protein 2, expressed on neutrophils, is involved in lipid accumulation, and lipid mediators like prostaglandin E2 can impair T cell functions [24]. Moreover, cholesterol efflux in macrophages can drive TAM-mediated tumor progression, including inhibition of IFN-γ induced gene expression [25]. These lipid-driven alterations not only support tumor cell energy demands but also foster a pro-tumor immune phenotype within the TME [24,25]. In UTUC, increased neutrophils correlate with malignant progression and suppression of lymphocyte-mediated anti-tumor responses [11,12]. While tumor-associated neutrophils in UTUC have been primarily studied in the context of NLR and clinical outcomes, the underlying molecular mechanisms remain largely unexplored.

In this study, we investigate the role of tumor-derived factors in modulating neutrophil function in UTUC. Using a proteomic array approach, we identified Apolipoprotein A1 (Apo-A1) as a secreted factor from UTUC tumors that exerts a modulatory effect on neutrophils. We found that Apo-A1 upregulated the expression of arginase-1 in neutrophils. This upregulation was mediated through signal transducer and activator of transcription-3 (STAT3) activation. Importantly, blocking the interactions between Apo-A1 and its receptors abolished the immunosuppressive functions of neutrophils. Additionally, we analyzed the phenotypic and functional characteristics of peripheral neutrophils, as well as the NLR values in UTUC patients.

## 2. Materials and Methods

### 2.1. Study Participants

This study enrolled participants from regular urological practices at the Chia-Yi Christian Hospital. The tumor lesions in the renal pelvis and ureter were confirmed by computed tomography. The diagnosis of UTUC was confirmed by pathological evidence from endoscopic biopsies or surgical resection of urinary tract cancers. Table 1 presents hematological data, including proportions of peripheral neutrophil and lymphocyte, as well as the characteristics of the study participants. Ethical approval was obtained from the Chia-Yi Christian Hospital Ethics Committee (No. 2020121), and the study was conducted in accordance with the ethical principles outlined in the Declaration of Helsinki for medical research involving human subjects. Participants gave informed consent to participate in the study before taking part.

### 2.2. Tumor Tissue Culture Supernatant (TTCS) and Tumor-Infiltrating Cell Collection

Before collecting the TTCS, surgical isolated tumor tissue was washed with 1× PBS for elimination of residual mucus and blood. The tissues were minced into small pieces using sterile scissors and subsequently plated in RPMI1640 medium supplemented with penicillin (100 units/mL, HIMEDIA, Mumbai, India), streptomycin (100 μg/mL, HIMEDIA, Mumbai, India), and 10% fetal bovine serum (FBS, Gibco, Grand Island, NY, USA). The culture was incubated for 24 h using 4 mL of medium per gram of tumor, and the supernatant was collected after centrifugation. To collect the tumor-infiltrating cells, tumor tissues were digested with collagenase D and DNase I, then passed through 100 μm cell strainers (BD Falcon, Franklin Lakes, NJ, USA). After resuspending the cells in culture medium, Ficoll-Paque gradient centrifugation was performed. Cells at the interface were collected and washed with 1× PBS, and the cell count was determined after replacing PBS with culture medium.

### 2.3. Protein Array

The collected TTCS and control medium were analyzed using a Human XL Cytokine Array Kit (R&D Systems, Minneapolis, MN, USA). A total of 1.5 mL array buffer containing 250 μL TTCS or control medium was incubated with an array membrane overnight at 4 °C. The detection antibody cocktails were then added to the array membrane according to the manufacturer’s protocol. Cytokine dots were performed by chemiluminescence detection in the MultiGel-21 imaging system (TOPBIO, Taipei, Taiwan). Quantitative data from images of the spot pixel density of cytokine array was performed by using MATLAB software (version 9.6).

### 2.4. In Vitro Neutrophil Culture and Neutrophil/T Cell Coculture

Fresh neutrophils were isolated from the donors (purity > 99%, Appendix A) using a human whole blood neutrophil isolation kit (Biolegend, San Diego, CA, USA) and were resuspended in RPMI1640 medium supplemented with 100 unit/mL penicillin, 100 μg/mL streptomycin, and 10% FBS. Neutrophils were treated with 20% TTCS or 40 μg/mL Apolipoprotein A1 (Apo-A1, PROSPEC, Ness-Ziona, Israel) in 5% CO_2_ at 37 °C. One hour later, neutrophils were washed with 1× PBS twice and dissolved in Trizol reagent (Ambion, Carlsbad, CA, USA) for total RNA isolation. For phosphorylated STAT3 (pSTAT3) or arginase-1 expression, neutrophils were stained with pSTAT3 antibody (Biolegend, San Diego, CA, USA) or arginase-1 antibody (Biolegend, San Diego, CA, USA), respectively. In blocking experiments associated with Apo-A1 signaling, neutrophils were preincubated with anti-SRB1 (reaction concentration 1:200, Novus Biologicals, Littleton, CO, USA), anti-ABCA1 (1:200, Novus Biologicals, Littleton, CO, USA), or anti-SRB1 and anti-ABCA1 Abs simultaneously for 30 min before they encountered the TTCS or Apo-A1. The STAT3-specific inhibitor JSI-124 (cucurbitacin I, Sigma-Aldrich, St. Louis, MO, USA) was preincubated with neutrophils for 10 min before TTCS or Apo-A1 treatment. To compare the neutrophil function in T cell proliferation between UTUC patients and healthy subjects, purified neutrophils from two groups were cocultured with T cells from the donors.

CD4^+^ or CD8^+^ T cells from PBMCs of the donors were performed by negative selection using a human T lymphocyte enrichment kit (BD Bioscience, San Jose, CA, USA). Neutrophils were cocultured with carboxyfluorescein succinimidyl ester (CFSE, ThermoFisher, Eugene, OR, USA)-labeled CD4^+^ or CD8^+^ T cells (neutrophil:T = 1:1), in 96-well plates. The coculture wells were incubated with anti-CD3/CD28 Abs (Dynabeads^TM^ human T activator CD3/CD28, ThermoFisher, Waltham, MA, USA) for T cell stimulation. Four days after coculture, CFSE-diluted signal was assessed for determination of proliferative T cell population.

### 2.5. RNA Extraction and Quantitative Real-Time PCR

The total RNA of the neutrophil was isolated using a GENEzol^TM^ TriRNA Pure Kit (Geneaid, Taipei, Taiwan) according to the manufacturer’s protocol. cDNA was synthesized from 0.5 μg RNA using an MMLV Reverse Transcription Kit (Protech Technology Enterprise, Taipei, Taiwan). Quantitative real-time PCR with an SYBR Green on StepOne cycler (Applied Biosystems/Life Technologies, Foster City, CA, USA) was used to amplify genes. Specific primers for the *CD274* gene included F: 5′-TCACTTGGTAATTCTGGGAGC-3′ and R: 5′-CTTTGAGTTTGTATCTTGGATGCC-3′; for the *ARG1* gene, they included F: 5′-GGCAAGGTGATGGAAGAAAC-3′ and R: 5′-AGTCCGAAACAAGCCAAGGT-3′; for the *ACTB* gene, they included F: 5′-TGCGTGACATTAAGGAGAAG-3′ and R: 5′-GCTCGTAGCTCTTCTCCA-3′. The expression of the target gene was determined relative to that of β-actin, and the relative fold change was calculated by the ΔΔCt method.

### 2.6. Flow Cytometry

The cells were resuspended with staining buffer (1× PBS containing 2% FBS and 2 mM EDTA) and then stained with fluorescent dye-conjugated Ab at 4 °C, for 30 min in the dark. For intracellular staining, the cells were fixed and permeabilized by Cytofix/Cytoperm buffer (BD Biosciences, CA, USA) for 15 min. The cells were followed by staining with fluorescent dye-conjugated Ab at 4 °C for 30 min in the dark. Before the pSTAT3-specific Ab staining, cells were permeabilized by Phosflow^TM^ perm buffer III (BD Biosciences, CA, USA) for 30 min. Accuri C6 plus flow cytometer (BD Biosciences, CA, USA) was used for evaluating the cell markers, and the data were further analyzed by FlowJo software (version 10). Mouse IgG1-PE (clone P3.6.2.8.1) and anti-human CD11b-PE (ICRF44) were purchased from Thermo Fisher. Mouse IgG2b-APC (clone MPC-11), anti-human CD66b-FITC (G10F5), CD15-FITC (SSEA-1), CD66b-PE (6/40c), pSTAT3-PE (13A3-1), CD45-PECy7 (2D1), CD15-APC (SSEA-1), CD4-APC (RPA-T4), and arginase-1-APC (14D2C43) were purchased from Biolegend. Anti-human CD8-FITC (clone RPA-T8), CD3-PE (UCHT1), PD-L1-PE (M1H1), and 7-AAD were purchased from BD Biosciences.

### 2.7. Enzyme-Linked Immunosorbent Assay (ELISA) and Cytometric Bead Array (CBA)

The collected TTCS from UTUC patients was assayed for Apo-A1 expression, using an ELISA kit (Abcam, Cambridge, MA, USA) according to the manufacturer’s instructions. For assessing the cytokine level in supernatants from coculture experiments, the CBA kit (BD Biosciences, CA, USA) including Abs for targeting IL-2, IL-4, IL-6, IL-10, IFN-γ, IL-17A, and TNF was used according to the manufacturer’s recommendation and analyzed by flow cytometry.

### 2.8. Statistical Analysis

Statistical analysis was performed using GraphPad Prism version 7 (GraphPad Software Inc., San Diego, CA, USA). Statistical comparisons were performed using an unpaired *t*-test to compare the means between two groups, and the Mann–Whitney test for non-normally distributed data. Spearman’s rank-correlation test was used to determine the association between two markers. *p*-values < 0.05 were considered statistically significant. The study participants’ characteristics are analyzed as means ± standard deviations.

## 3. Results

### 3.1. UTUC Tumor-Derived Apo-A1 Increased Arginase-1 Expression in Neutrophils

The tumor microenvironment secretome not only induced chemotaxis of neutrophils but also orchestrated the phenotype of tumor-associated neutrophils [26,27]. To identify tumor-derived factors, the supernatant from UTUC biopsy tissues was collected and analyzed using a membrane-based antibody array (Figure 1A). The ten most upregulated factors in the TTCS were identified as angiogenin, adiponectin, apolipoprotein A1 (Apo-A1), macrophage migration inhibitory factor (MIF), EMMPRIN (extracellular matrix metalloproteinase inducer), MMP-9, growth differentiation factor 15 (GDF-15), lipocalin-2, vitamin D-binding protein, and endoglin (Figure 1B).

GM-CSF protein expression showed no significant difference between TTCS and the control medium. To investigate whether UTUC-derived supernatant modulates neutrophils, neutrophils were treated with 20% TTCS. The expression of arginase-1 (ARG1) was significantly increased (Figure 1C. *ARG1*, *p* = 0.003), whereas PD-L1 (CD274) expression remained unchanged (Figure 1C. *CD274*). Among the top three upregulated factors, angiogenin was found to inhibit neutrophil degranulation, promoting angiogenesis by suppressing angiostatin activity [28]. Adiponectin inhibited inflammatory cytokine and promoted M2 macrophage phenotype, including the upregulation of IL-10 and arginase-1, in macrophages [29,30]. Apo-A1, a well-known major structural protein of high-density lipoprotein (HDL), regulates cholesterol trafficking and affects immune responses [31]. Previous studies demonstrated that Apo-A1 treatment directly attenuates neutrophil activity in inflammatory conditions [32,33]. Additionally, HDL treatment was shown to increase arginase-1 and Fizz-1 expression in primary murine macrophages [34].

In addition to the antibody array, tumor-derived factors were also analyzed via Western blot, which revealed a marked increase in Apo-A1 levels in TTCS from UTUC patients compared to the control medium (Appendix A). To determine which factors induced arginase-1 expression in UTUC-derived supernatant, adiponectin and Apo-A1 were evaluated. Quantitative real-time PCR showed that Apo-A1 significantly upregulated *ARG1* expression in neutrophils (Figure 1D. ApoA1 vs. ctrl, *p* < 0.001; TTCS vs. ctrl, *p* = 0.006), while adiponectin alone did not induce *ARG1* expression (Appendix A). These findings were corroborated by flow cytometry (Figure 1E. ApoA1 vs. ctrl and TTCS vs. ctrl, *p* < 0.001) and Western blot analyses (Appendix A. ApoA1 vs. ctrl and TTCS vs. ctrl, *p* = 0.001 and *p* = 0.021). In summary, our findings demonstrate that Apo-A1 is markedly elevated in the UTUC microenvironment and promotes arginase-1 expression in neutrophils.

### 3.2. STAT3 Activation Involved in Arginase-1 Upregulation of Neutrophils by Apo-A1 Signaling

The interactions of Apo-A1 with its receptors are well documented in lipid metabolism and are known to mediate both pro- or anti-inflammatory immune responses [35,36]. To explore whether Apo-A1 signaling regulates arginase-1 expression, neutralizing antibodies targeting Apo-A1 receptors, including anti-SRB1 (scavenger receptor class B type 1) and anti-ABCA1 (ATP binding cassette transporter A1) antibodies (Abs), were utilized. Pretreatment with either anti-SRB1 or anti-ABCA1 antibodies independently abolished Apo-A1-induced arginase-1 upregulation in neutrophils, both at the RNA level (Figure 2A. ApoA1 + anti-SRB1 vs. ApoA1, *p* = 0.005; ApoA1 + anti-ABCA1 vs. ApoA1, *p* < 0.001) and the protein level (Figure 2B. ApoA1 + anti-SRB1 vs. ApoA1, *p* = 0.004; ApoA1 + anti-ABCA1 vs. ApoA1, *p* < 0.001).

Simultaneous blockade of SRB1 and ABCA1 receptors further suppressed arginase-1 expression in neutrophils treated with TTCS, as shown by significant reductions at both the RNA level (Figure 2C. TTCS + blockers vs. TTCS, *p* < 0.001) and protein level (by flow cytometry: Figure 2D. TTCS + blockers vs. TTCS, *p* < 0.001; by Western blot: Appendix A. TTCS + blockers vs. TTCS, *p* = 0.008). Similar results were observed in Apo-A1-treated neutrophils, with arginase-1 protein expression significantly reduced by receptor blockade (Appendix A. ApoA1 + blockers vs. ApoA1, *p* = 0.001). These findings demonstrate that Apo-A1-induced arginase-1 expression in neutrophils can be effectively reversed by neutralizing antibodies targeting SRB1 and ABCA1, highlighting the critical role of these receptors in Apo-A1 signaling to induce arginase-1 expression in neutrophils.

The Apo-A1 induced signal transduction via the STAT3 pathway has been correlated with anti-inflammatory response in macrophages [35]. To explore whether STAT3 activation is involved in TTCS-treated neutrophils, we analyzed the phosphorylation of STAT3 (pSTAT3) in response to Apo-A1 and TTCS treatment. Our findings revealed that both Apo-A1 and TTCS treatment significantly upregulated pSTAT3 expression in neutrophils (Figure 2E. left column; Figure 2F. ApoA1 vs. ctrl, *p* < 0.001; TTCS vs. ctrl, *p* = 0.005). When neutrophils were pretreated with the STAT3-specific inhibitor JSI-124 (Figure 2E. middle column), the increased pSTAT3 expression observed in the Apo-A1 and TTCS groups was significantly suppressed (Figure 2E,F. ApoA1 + JSI124 vs. ApoA1, *p* = 0.002; TTCS + JSI-124 vs. TTCS, *p* = 0.037). These results indicate that STAT3 phosphorylation is specifically induced by Apo-A1 and TTCS treatment. To further investigate whether STAT3 activation mediated by Apo-A1 receptor signaling was responsible for the effects observed in TTCS-treated neutrophils, neutralizing antibodies targeting Apo-A1 receptors, including anti-SRB1 and anti-ABCA1, were utilized. Simultaneous blockade of SRB1 and ABCA1 receptors (Figure 2E. right column, blockers) significantly reversed the upregulation of pSTAT3 expression in neutrophils (Figure 2E,F. ApoA1 + blockers vs. ApoA1, *p* < 0.001; TTCS + blockers vs. TTCS, *p* = 0.005). To investigate whether STAT3 activation is responsible for arginase-1 upregulation, neutrophils were pretreated with the STAT3-specific inhibitor JSI-124 prior to Apo-A1 or TTCS exposure. As anticipated, pretreatment with JSI-124 effectively abrogated arginase-1 expression in neutrophils induced by either Apo-A1 or TTCS (Figure 2G. ApoA1 + JSI124 vs. ApoA1 and TTCS + JSI-124 vs. TTCS, *p* < 0.001). The above results showed that STAT3 activation was involved in arginase-1 upregulation of neutrophils treated by Apo-A1 and TTCS.

### 3.3. Apo-A1 Signaling in Neutrophils Involved in UTUC Mediated T Cell Suppression

To determine whether the modulation of T cell functions by neutrophils in UTUC depends on Apo-A1 in TTCS, neutrophils pretreated with either Apo-A1 or TTCS were cocultured with T cells. Apo-A1- or TTCS-pretreated neutrophils reduced both the CD4^+^ T (Figure 3A,B. ApoA1 vs. ctrl, *p* = 0.006; TTCS vs. ctrl, *p* < 0.001) and CD8^+^ T cell proliferation (Figure 3C,D. ApoA1 vs. ctrl, *p* = 0.003; TTCS vs. ctrl, *p* = 0.002).

Given that the interaction between Apo-A1 and its receptors is responsible for arginase-1 induction in neutrophils (Figure 2B,C), which likely affects T cell activity, neutralizing Abs targeting Apo-A1 receptors on neutrophils were employed to assess their role in modulating T cell responses. Blocking Apo-A1 receptors with these antibodies effectively reversed the inhibitory effects on CD4+ T cell proliferation (Figure 3A,B. ApoA1 + blockers vs. ApoA1, *p* = 0.015; TTCS + blockers vs. TTCS, *p* = 0.002) and CD8^+^ T cells proliferation (Figure 3C,D. ApoA1 + blockers vs. ApoA1, *p* = 0.001; TTCS + blockers vs. TTCS, *p* = 0.017). These findings demonstrate that Apo-A1 secreted by UTUC contributes to neutrophil-mediated suppression of T cell proliferation.

### 3.4. The Characteristics of Neutrophils in UTUC Patients

Suppressive circulating neutrophils, identified as a lower-density population within PBMCs, express typical granulocyte markers such as CD66b and CD15 and have been associated with cancer progression [15,37]. In this study, 20 UTUC patients and 21 age-matched healthy subjects were enrolled to evaluate neutrophil characteristics within PBMCs. The proportion of CD66b^+^CD15^+^ neutrophils was significantly increased in UTUC patients compared to healthy subjects (Figure 4A. *p* < 0.001). The NLR of UTUC patients was higher than that of healthy subjects (Table 1).

Interestingly, the expression of the checkpoint inhibitor PD-L1 on neutrophils did not differ between UTUC patients and healthy subjects (Figure 4B). However, arginase-1 expression in neutrophils was significantly elevated in UTUC patients (Figure 4C. *p* = 0.002). In contrast, the proportion of CD4^+^ T cells in UTUC patients was significantly lower than in healthy subjects, while the percentages of CD3^+^ and CD8^+^ T cells remained unchanged (Figure 4D–F. *p* = 0.004 in CD4 T). These results indicated a higher proportion of neutrophils with elevated arginase-1 expression and a decreased CD4^+^ T cell population in the peripheral blood of UTUC patients.

Next, tumor-infiltrating cells from UTUC patients were analyzed for arginase-1 expression in neutrophils and T cell populations. A negative correlation was observed between the proportion of infiltrating CD4^+^ T cells and arginase-1 expression in tumor-infiltrating neutrophils (Figure 5A,B). Furthermore, Apo-A1 levels in TTCS inversely correlated with infiltrating CD4^+^ T cells but not CD8^+^ T cells (Figure 5C,D). These results highlight the interplay between neutrophils with elevated arginase-1 expression and the suppression of CD4^+^ T cell populations in both peripheral blood and tumor microenvironments of UTUC patients.

To evaluate the impact of neutrophils on CD4^+^ T cell proliferation in UTUC, neutrophils isolated from whole blood were cocultured with CD4^+^ T cells purified from healthy donors. The results demonstrated that neutrophils from UTUC patients significantly inhibited CD4^+^ T cell proliferation (Figure 6A,B. *p* < 0.001). Furthermore, the secreted cytokines, including IL-2 and IFN-γ, was significantly reduced when CD4^+^ T cells were cocultured with neutrophils from UTUC patients (Figure 6C. IL-2, *p* = 0.012; IFN-γ, *p* = 0.04). These findings indicate that neutrophils from UTUC patients exhibit pronounced immunosuppressive effects on T cells.

## 4. Discussion

This study reveals that UTUC modulates the immunosuppressive function of neutrophils via Apo-A1/pSTAT3 axis-induced arginase-1 expression, and interruption of the binding between Apo-A1 and its receptors on neutrophils abrogates the suppressive effect on T cells. Following the in vitro experiments, neutrophils treated with primary biopsy tumor tissue supernatants exhibited increased arginase-1 expression. The Apo-A1 level in TTCS or arginase-1 expression in tumor-infiltrating neutrophils is negatively correlated with tumor-infiltrating CD4^+^ T population. Functionally, UTUC-associated neutrophils inhibit CD4+ T cell proliferation and cytokine production, including IL-2 and IFN-γ. These findings indicate that UTUC leverages Apo-A1 to induce arginase-1 expression in neutrophils, thereby suppressing T cell activity.

It has been suggested that circulating G-MDSCs predominantly increase in various cancers, playing an essential role in cancer progression [38]. Similarly, circulating neutrophils often exhibit immunosuppressive phenotypes, infiltrating into tumor tissues and promoting progression and metastasis [16]. The pro-tumor neutrophils share phenotypic and functional similarities with G-MDSCs [38]. For instance, bladder tumors induce the PD-L1 expression on stromal neutrophils to cause T cell inhibition [18]. However, contrasting roles of neutrophils have been reported, as tumor-associated neutrophils are enriched in the “hot tumor” basal-type bladder cancer and are associated with a better outcome [39]. These findings underscore the dual roles of neutrophils in cancer. Our present study highlights a pro-tumor role of neutrophils in UTUC through arginase-1 upregulation, contributing to T cell suppression.

Arginase-1 is mainly expressed in the liver and myeloid lineage cells. The physiological functions of arginase-1, through the hydrolysis of arginine, perform ammonia detoxification and the generation of downstream metabolites for cell development and collagen formation [21,40]. Arginine is crucial for cellular metabolism, promoting T cell survival and function. [41]. Depletion of extracellular arginine pool by arginase-1 results in a reduction in available arginine for effector T cells, leading to T cell suppression [22]. In lung adenocarcinoma tissues, arginase-1 localizes predominantly to CD66b+ neutrophils and inversely correlates with CD3^+^ T cells [42]. Tumor-educated neutrophils and monocytes acquire immunosuppressive functions, including arginase-1 expression and anti-inflammatory cytokine production [43]. Increased arginase-1 expression is a poor prognostic factor in various cancer types [22]. Arginase inhibitors, combined with immune checkpoint inhibitors, are under clinical development as therapeutic strategies. [21]. In the present study, we demonstrate that UTUC induces arginase-1 expression in neutrophils to associate with T cell suppression, further implicating its role in immune evasion.

The Apo-A1/ABCA1 interaction promotes signal transduction and cholesterol efflux activity [35]. Apo-A1/ABCA1-STAT3 axis inhibits LPS-induced IL-6, IL-1β, and TNFα production by macrophages [35], suggesting the anti-inflammatory role of Apo-A1/ABCA1 signaling. It has been revealed that cholesterol efflux induced by Apo-A1/ABCA1 interaction enhanced IL-4 signaling, significantly increasing arginase-1 expression in TAMs [25]. This phenomenon is associated with immunosuppression in the TME [25]. While Apo-A1 has varying prognostic implications in different cancers, evidence links its overexpression with cancer progression, including increased serum levels in hepatocellular carcinoma [44]. Moreover, the increased Apo-A1 protein from urine has been suggested as a potential biomarker of bladder cancer [45]. We previously provided evidence of a correlation between serum Apo-A1 levels and the presence of infiltrating neutrophils and T lymphocytes in UTUC tumors [46]. Interestingly, infiltrating immune cells show an inverse correlation with Apo-A1 expression in the tumor tissue of renal clear cell carcinoma [47]. Our present study demonstrates that Apo-A1 is increased in TTCS, further inducing arginase-1 upregulation in neutrophils of UTUC. The blockade of Apo-A1/ABCA1 interaction caused a reduction in arginase-1 expression in both mRNA and protein levels (Figure 2A,B), reflecting that ABCA1 induces arginase-1 expression by Apo-A1 treatment. In primary tumor tissue, infiltrating CD4^+^ T cells negatively correlated with Apo-A1 protein level and arginase-1 expression in infiltrating neutrophils (Figure 5A,C). This indicates that Apo-A1 signaling via ABCA1 is integral to neutrophil-mediated immunosuppression in UTUC.

SRB1 and CD36 (SRB2) belong to the class B family of scavenger receptors and participate in lipid metabolism [36]. CD36, for example, facilitates metabolic signaling in cancer cells and TAMs, promoting tumor growth [17]. In cancer cells, higher glucose uptake increases the SRB1 expression, promoting HDL/SRB1 binding and increasing cholesterol influx, contributing to proliferation and migration [48]. Several recent studies have reported that highly expressed SRB1 in tumor tissues is associated with poor prognosis [49,50,51]. In lymphocytes and dendritic cells, HDL/SRB1 binding-induced cholesterol efflux causes a reduction in antigen presentation and cell differentiation [52]. In this study, the interaction of Apo-A1 with SRB1 similarly contributes to arginase-1 upregulation in UTUC (Figure 2A,B). Blocking both ABCA1 and SRB1 receptors abrogated TTCS-induced arginase-1 expression (Figure 2D), further highlighting the immunosuppressive role of Apo-A1 signaling in neutrophils.

Enhanced STAT3 activity plays a critical role in tumor development, affecting both tumor and immune cells [53]. For example, cancer cell-intrinsic STAT3 activation fosters secretomes that recruit and polarize neutrophils [54], while silencing STAT3 in G-MDSCs restores T cell proliferation and cytokine secretion [55]. Lung cancer has been examined to drive neutrophils toward an N2-like phenotype through STAT3 hyperactivation to enhance cancer cell migration [56]. This accumulating evidence suggests the pro-tumor function of STAT3 activation in neutrophils. In this study, we demonstrate that blockage of Apo-A1 signaling reduced the STAT3 activation in UTUC-treated neutrophils (Figure 2E,F), and thus reversed the suppressive effect on the T cells (Figure 3), demonstrating the mediator of the STAT3 molecule in UTUC-orchestrated neutrophils.

Tumor-associated neutrophils and macrophages could produce numerous proteases, including MMPs, which contribute to extracellular matrix degradation, thereby supporting tumor development and metastasis [57]. EMMPRIN, a well-known inducer of MMPs, expresses in various tissue and regulates physiological and pathological processes [58]. Overexpression of EMMPRIN in tumor tissues has been associated with poor prognosis in bladder cancer [58] and is considered a potential prognostic factor in various other cancers [59]. EMMPRIN expression on the surface of neutrophils enhances MMP production and chemotaxis, linking it to the pathogenesis of inflammatory diseases [60]. In many solid tumors, MMPs produced by stromal cells facilitate the release of soluble EMMPRIN from the plasma membrane of tumor cells, creating a positive feedback loop that further induces MMP expression [58]. In this study, among the previously mentioned top three upregulated factors in UTUC, we focused on examining the role of Apo-A1 in immunosuppressive neutrophils, rather than investigating the other identified factors, including EMMPRIN and MMP-9. Therefore, the specific role of neutrophils in cancer metastasis and tumor development was not explored in our study.

## 5. Conclusions

In summary, our study demonstrates that UTUC enhances arginase-1 expression in neutrophils through the Apo-A1/STAT3 axis, thus inhibiting T cell functions (Figure 7). These findings provide insight into mechanisms in immunosuppression of neutrophils in UTUC and suggest potential therapeutic targets.

## Figures and Tables

**Figure 1 cells-14-00660-f001:**
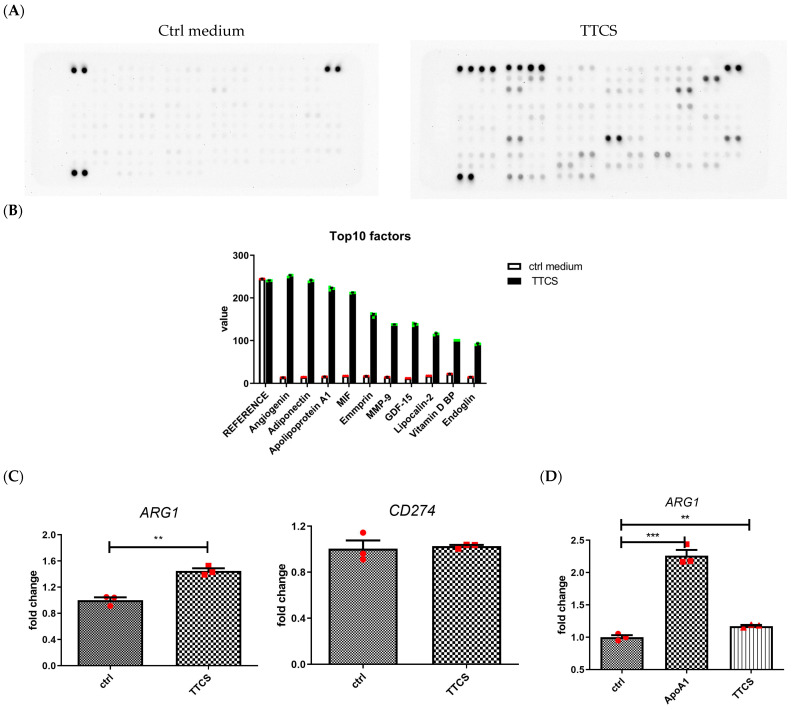
Arginase-1 expression was increased in neutrophils treated by UTUC tumor tissue cultured supernatant (TTCS). (**A**) Proteome Profiler Cytokine Arrays of TTCS (right panel) or control medium (ctrl medium, left panel) are presented. (**B**) Top ten upregulated proteins from the above protein array were measured by image analysis software. (**C**) The RNA expression level of arginase-1 (**left panel**) and PD-L1 (**right panel**) were measured in primary neutrophils treated with TTCS or control medium (ctrl). (**D**,**E**) Primary neutrophils were treated with apolipoprotein A1 (Apo-A1), TTCS, or control medium. The RNA expression level of arginase-1 was evaluated (**D**). The protein expression level of arginase-1 is shown in a histogram (**E**, **left panel**), and the corresponding statistical analysis is presented (**E**, **right panel**). Three independent experiments were performed. ** *p* < 0.01 and *** *p* < 0.001, unpaired *t*-tests.

**Figure 2 cells-14-00660-f002:**
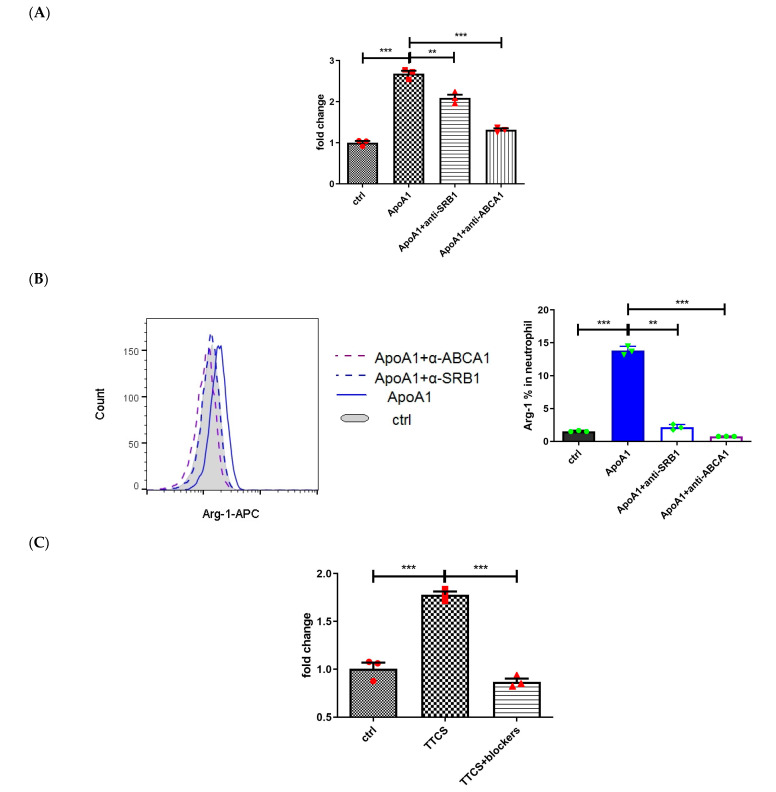
Apo-A1/pSTAT3 was responsible for arginase-1 upregulation in neutrophils treated by TTCS of UTUC. (**A**–**D**) Neutrophils were preincubated with Apo-A1 receptor antibodies, including anti-SRB1, anti-ABCA1 blocking Ab, or a combination of both two antibodies (TTCS + blockers) for 30 min. Subsequently, they were treated with Apo-A1 (**A**,**B**) or TTCS (**C**,**D**). The RNA expression (**A**,**C**) and protein levels (**B**,**D**) of arginase-1 in neutrophils are presented in the left panel, with corresponding statistical analysis displayed in the right panel. (**E**) The dot plot illustrates the pSTAT3 expression of neutrophils preincubated with STAT3-specific inhibitor (JSI-124) or Apo-A1 blockers (α-SRB1 + α-ABCA1), followed by treatment of Apo-A1, TTCS, or control medium (ctrl). The statistical analysis data of (**E**) are presented in (**F**). (**G**) The histogram of arginase-1 protein of neutrophils treated with Apo-A1 (blue line), TTCS (red line), or control (filled gray) in combination with STAT3-specific inhibitors (dotted line) is represented in the left panel, and the statistical analysis is shown in the right panel. Two independent experiments were performed. * *p* < 0.05, ** *p* < 0.01, and *** *p* < 0.001, unpaired *t*-tests.

**Figure 3 cells-14-00660-f003:**
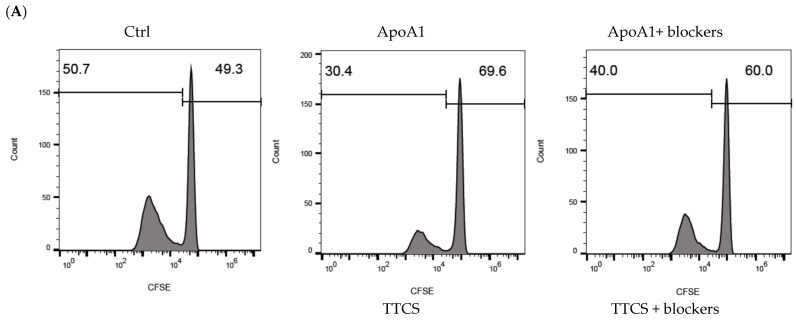
Apo-A1 signaling in neutrophils involved in suppression of CD4 and CD8^+^ T cell proliferation treated by TTCS of UTUC. Neutrophils from the donors were preincubated with anti-SRB1 and anti-ABCA1 Abs simultaneously or with PBS for 30 min. Subsequently, they were treated with TTCS, Apo-A1, or control medium (ctrl). After that, neutrophils were cocultured with carboxyfluorescein succinimidyl ester (CFSE)-labeled CD4^+^ T cells (**A**,**B**) or CD8^+^ T cells (**C**,**D**) at a ratio of 1:1. The histograms display the proliferation of CD4^+^ T cells (**A**) and CD8^+^ T cells (**C**), and the statistical results for CD4^+^ T cells (**B**) and CD8^+^ T cells (**D**) are presented. One of three independent experiments is represented. * *p* < 0.05, ** *p* < 0.01, and *** *p* < 0.001, unpaired *t*-tests.

**Figure 4 cells-14-00660-f004:**
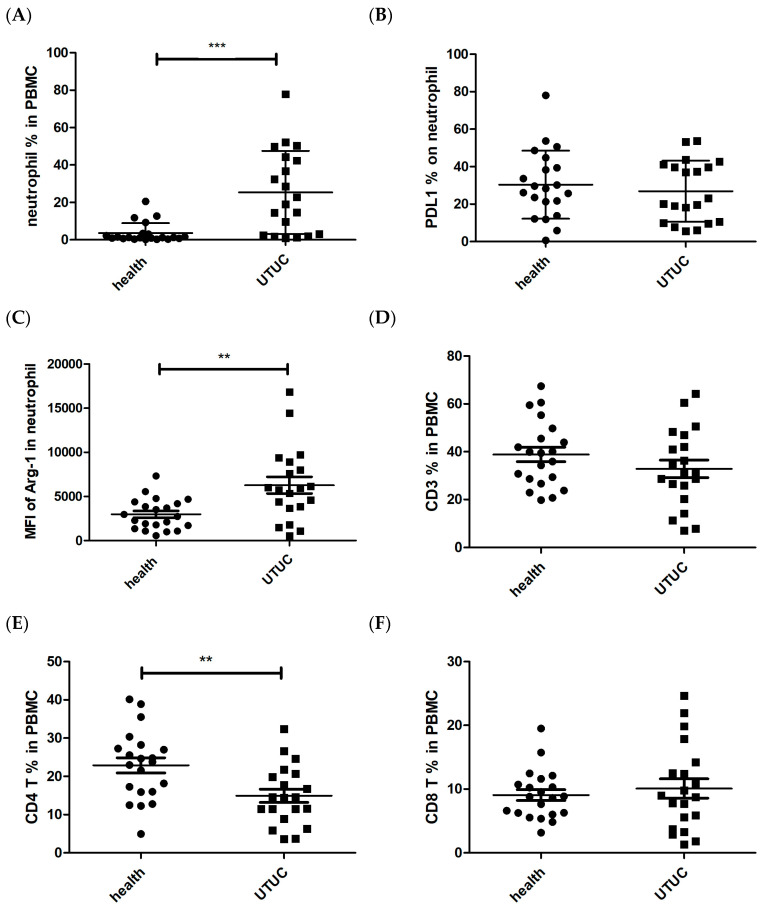
The characteristics of neutrophils and T cells in peripheral blood mononuclear cells (PBMCs) and tumors of UTUC patients. (**A**) The percentage of CD66b+ CD15+ neutrophil population of PBMC, (**B**) the percentage of PD-L1 expression on neutrophils, (**C**) the mean fluorescence intensity of arginase-1 in neutrophils, (**D**) the percentage of CD3^+^ population of PBMC, (**E**) the percentage of CD3^+^CD4^+^ T cell population of PBMCs, and (**F**) the percentage of CD3^+^CD8^+^ T cell population of PBMC. Peripheral blood of UTUC patients (*n* = 20) and healthy subjects (*n* = 21) were collected and analyzed by flow cytometry. ** *p* < 0.01 and *** *p* < 0.001. Statistical comparisons of (**A**,**B**) were analyzed using Mann–Whitney test, and others were analyzed using unpaired *t*-tests.

**Figure 5 cells-14-00660-f005:**
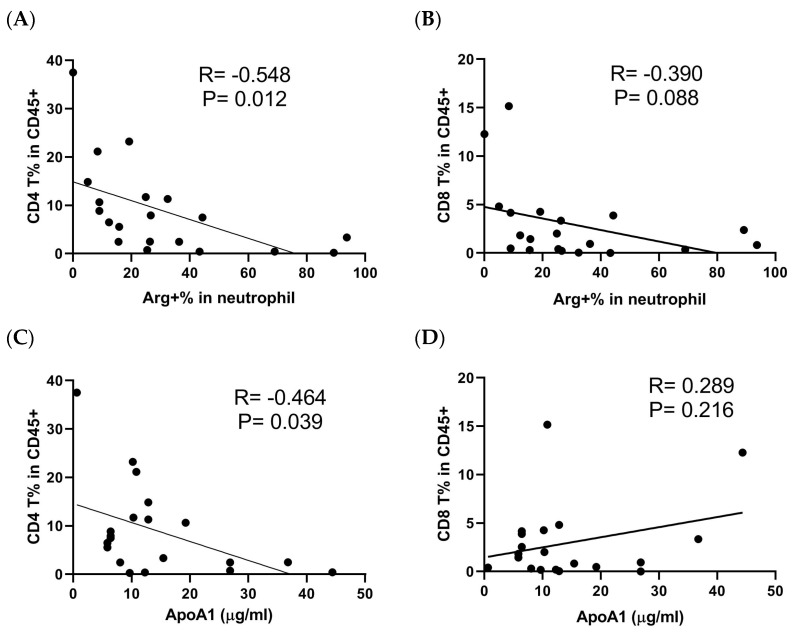
Assessment of the cell populations, including infiltrating neutrophils, T lymphocytes, and the Apo-A1 protein level in UTUC tumor tissue. The tumor-infiltrating cells isolated from UTUC patients were assessed by flow cytometry, and the percentage of (**A**) CD4^+^ T (CD4^+^CD3^+^) or (**B**) CD8^+^ T (CD8^+^CD3^+^) within infiltrating CD45^+^ cells was correlated with the arginase-1 expression in infiltrating neutrophils. The Apo-A1 protein level (μg/mL) in TTCS, assessed by ELISA, was correlated with the percentage of (**C**) CD4^+^ T or (**D**) CD8^+^ T population in tumor-infiltrating CD45^+^ cells. The Spearman method for rank correlation was applied to analyze the link between the two markers.

**Figure 6 cells-14-00660-f006:**
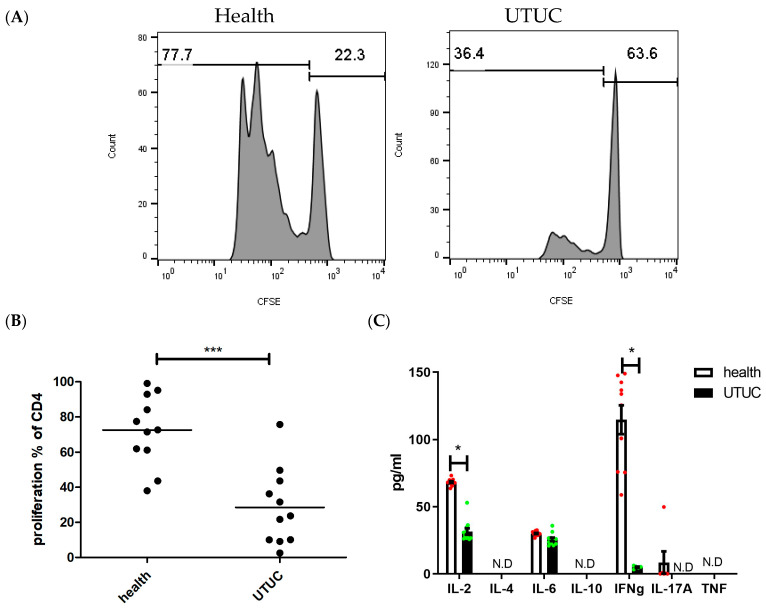
CD4^+^ T cell functions were inhibited by neutrophils from UTUC patients. CD4+ T cells were cocultured with peripheral blood neutrophils isolated from UTUC patients (UTUC) and healthy subjects (health). (**A**) Representative CD4^+^ T cell proliferations were analyzed by flow cytometry. (**B**) The corresponding statistical analysis is presented. Additionally, cytokines in the cocultured supernatants from UTUC patients (black bar) or healthy subjects (white bar) were assessed (**C**). * *p* < 0.05 and *** *p* < 0.001, unpaired *t*-tests (*n* = 11).

**Figure 7 cells-14-00660-f007:**
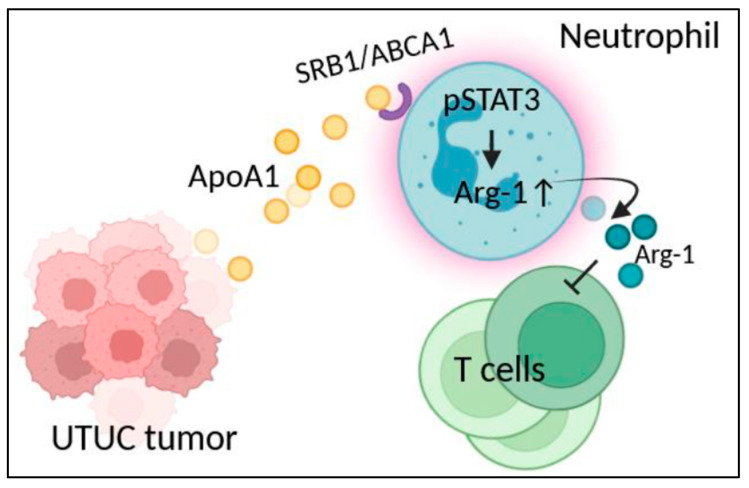
Summarized diagrams depict that UTUC enhances the arginase-1 expression in neutrophils through the Apo-A1/STAT3 axis to inhibit T cell functions. In the TME of UTUC, increased Apo-A1 protein binds to its receptors, such as SRB1 and ABCA1, on the surface of neutrophils. This binding induces a cell-intrinsic mechanism that activates STAT3. Phosphorylation of STAT3 then promotes arginase-1 expression in neutrophils, which leads to T cell suppression. These findings suggest that neutrophils play a key role as mediators in the immunosuppression orchestrated in UTUC. This figure was Created in Biorender. Chia-Bin Chang. (2023) https://www.biorender.com/ (accessed on 14 May 2023). https://app.biorender.com/illustrations/6481a41e5b09b6019c74cc3b (accesed on 14 May 2023).

**Table 1 cells-14-00660-t001:** Characteristics of the study participants. * *p* < 0.05, compared with healthy controls.

	UTUC Patients (*n* = 20)	Healthy Subjects (*n* = 21)
Age, year (mean ± SD)	67.65 ± 12.51	69.19 ± 5.14
Gender, *n* (%)		
Male	10 (50)	11 (52.2)
Female	10 (50)	10 (47.8)
Tumor site, *n* (%)		
Ureter	12 (60)	
Renal pelvic	11 (55)	
Both	3 (15)	
T stage, *n* (%)		
≤pT2	9 (45)	
>pT2	11 (55)	
Neutrophils, %	* 71.55 ± 9.15	63.97 ± 8.40
Lymphocytes, %	21.59 ± 9.22	26.73 ± 8.69
NLR	* 4.46 ± 3.52	2.69 ± 1.53

## Data Availability

Data are contained within the article and Appendix A.

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
