# Peer review of "Immunosuppression of Tumor-Derived Factors Modulated Neutrophils in Upper Tract Urothelial Carcinoma Through Upregulation of Arginase-1 via ApoA1-STAT3 Axis"

_cells, 2025, doi:10.3390/cells14090660_

Round 1
Reviewer 1 Report
Comments and Suggestions for Authors
Manuscript titled “Immunosuppression of tumor-derived factors modulated neutrophils in upper tract urothelial carcinoma through upregulation of arginase-1 via ApoA1-STAT3 axis” focuses on how tumor derived factors modulate neutrophils and consecutively T cell responses in upper tract urothelial carcinoma (UTUC). They found that tumor secretome protein ApoA1 plays a crucial role in upregulation of Arginase-1 expression level in neutrophil. This increase in Arginase-1 level expression in neutrophil is mediated by STAT3 signaling, evidenced by increase in pSTAT3 expression. Blocking interaction between ApoA1 protein and its receptor leads to reduced Arginase-1 expression in neutrophils. Arginase-1 high neutrophils promote decrease in CD4+and CD8+ T cell proliferation and blocking ApoA1 interaction with its receptor reverses it. The authors also focus on the correlation between ApoA1 level, Arginase1 expression, and T cell infilitration in patient samples.
Overall, this is a well-executed study that focuses on timely and relevant topic. It mostly cites relevant literature and incorporates proper control for the conducted experiments. However, I have several suggestions I would like to bring forward to improve the presentation of the figures.
- The authors should show data points for all the bar graphs, instead of just plotting mean and standard deviation
- Some of the figure panels quality in terms of resolution are not sharp, such as Figure 1A, 2E, 3A, 3C
- It might helpful for the reader if the authors can provide an illustration panel that summarizes how the signaling works at the end of the paper as part of final figure.

Author Response
Dear Editor and Reviewers:
We greatly thank you for the extensive assessment of our manuscript (cells-3601956), and for your invaluable comments. We have revised the original manuscript in accordance with reviewer’s comments, and corresponding changes are highlighted in red. Additionally, some sentences have been rewritten (highlighted in green) to reduce similarity with previously published papers, according to the editor’s suggestion. The Part I presents our responses (highlighted in blue) to reviewer’s comments (underline), while Part II outlines the revisions made in response to the editor’s suggestions or corrections of identified issues (highlighted in orange).
Part I:
Reviewer#1
Comments and Suggestions for Authors
Manuscript titled “Immunosuppression of tumor-derived factors modulated neutrophils in upper tract urothelial carcinoma through upregulation of arginase-1 via ApoA1-STAT3 axis” focuses on how tumor derived factors modulate neutrophils and consecutively T cell responses in upper tract urothelial carcinoma (UTUC). They found that tumor secretome protein ApoA1 plays a crucial role in upregulation of Arginase-1 expression level in neutrophil. This increase in Arginase-1 level expression in neutrophil is mediated by STAT3 signaling, evidenced by increase in pSTAT3 expression. Blocking interaction between ApoA1 protein and its receptor leads to reduced Arginase-1 expression in neutrophils. Arginase-1 high neutrophils promote decrease in CD4+and CD8+ T cell proliferation and blocking ApoA1 interaction with its receptor reverses it. The authors also focus on the correlation between ApoA1 level, Arginase1 expression, and T cell infilitration in patient samples.
Overall, this is a well-executed study that focuses on timely and relevant topic. It mostly cites relevant literature and incorporates proper control for the conducted experiments. However, I have several suggestions I would like to bring forward to improve the presentation of the figures.
- The authors should show data points for all the bar graphs, instead of just plotting mean and standard deviation
- Some of the figure panels quality in terms of resolution are not sharp, such as Figure 1A, 2E, 3A, 3C
- Responses:
In response to the reviewer’s suggestions, we have revised most of the figures to improve resolution and clarity. Additionally, we have updated all bar graphs to include the corresponding individual data points.
- It might helpful for the reader if the authors can provide an illustration panel that summarizes how the signaling works at the end of the paper as part of final figure.
- Responses:
In response to the reviewer’s suggestion, we have added the following sentences:
On Page19 (legend of Figure 7), lines 598-603
“In the TME of UTUC, increased Apo-A1 protein binds to its receptors, such as SRB1 and ABCA1, on the surface of neutrophils. This binding induces a cell-intrinsic mechanism that activates STAT3. Phosphorylation of STAT3 then promotes arginase-1 expression in neutrophils, which leads to T cell suppression. These findings suggest that neutrophils play a key role as mediators in the immunosuppression orchestrated in UTUC.”
Part II:
The revised words associated with editor’s suggestion or corrections of identified issues are described below:
On Page 3 (paragraph 2.1.), lines 121-122:
“The study enrolled participants from regular urological practices at the Chia-Yi Christian Hospital.” takes the place of this sentence “Study participants were recruited from customary urological practices at the Chia-Yi Christian Hospital.”
On Page 3 (paragraph 2.1.), lines 124-129:
“Table 1 presents hematological data, including proportions of peripheral neutrophil and lymphocyte, as well as the characteristics of the study participants. Ethical approval was obtained from the Chia-Yi Christian Hospital Ethics Committee (No. 2020121), and the study was conducted in accordance with the ethical principles outlined in the Declaration of Helsinki for medical research involving human subjects.” takes the place of these sentences “The hematological profiles provided the data on the proportion of neutrophils and lymphocytes in peripheral bloods of study participants (Table 1). This study was approved by the Ethics Committee of Chia-Yi Christian Hospital (No. 2020061) in Taiwan and followed the Declaration of Helsinki ethical principles for medical research involving human subjects.”
On Page 3 (paragraph 2.2.), lines 133-134:
“surgical isolated tumor tissue was washed with 1× PBS for elimination of residual mucus and blood” takes the place of this sentence “surgical isolated tumor tissue was washed with 1× PBS to remove blood and mucus”
On Page 3 (paragraph 2.2.), lines 135-138:
“RPMI1640 medium supplemented with penicillin (100 units/mL, HIMEDIA, Mumbai, India), streptomycin (100 μg/mL, HIMEDIA, Mumbai, India) and 10% fetal bovine serum (FBS, Gibco, NY, USA). The culture was incubated for 24 hours using 4 mL of medium per gram of tumor,” takes the place of this sentence “RPMI1640 medium (HIMEDIA, Mumbai, India) containing 10% fetal bovine serum (FBS, Gibco, NY, USA), 100 unit/mL penicillin (HIMEDIA, Mumbai, India), and 100 μg/mL streptomycin (HIMEDIA, Mumbai, India) for 24 hr (4 mL medium/ g tumor),”
On Page 3 (paragraph 2.2.), lines 141-144:
“After resuspending the cells in culture medium, Ficoll-Paque gradient centrifugation was performed.” takes the place of these sentences “The cells were resuspended with culture medium and subsequently subjected to Ficoll-Paque gradient centrifugation.”
On Page 4 (paragraph 2.4.), lines 157-158:
“RPMI1640 medium supplemented with 100 unit/mL penicillin, 100 μg/mL streptomycin and 10% FBS.” takes the place of this sentence “RPMI1640 medium containing 10% FBS, 100 unit/mL penicillin, and 100 μg/mL streptomycin.”
On Page 4 (paragraph 2.4.), lines 173-178:
“Neutrophils were cocultured with carboxyfluorescein succinimidyl ester (CFSE, ThermoFisher, Oregon, USA)-labeled CD4⁺ or CD8⁺ T cells (neutrophil: T=1:1), in 96 well plates. The coculture wells were incubated with anti-CD3/CD28 Abs (DynabeadsTM human T activator CD3/CD28, ThermoFisher, MA, USA) for T cell stimulation. Four days after coculture, CFSE-diluted signal was assessed for determination of proliferative T cell population.” takes the place of these sentences “Neutrophils were cocultured with carboxyfluorescein succinimidyl ester (CFSE, ThermoFisher, Oregon, USA)-labeled CD4⁺ or CD8⁺ T cells, at a ratio of 1:1, in 96 well plates, upon anti-CD3/CD28 stimulation (DynabeadsTM human T activator CD3/CD28, ThermoFisher, MA, USA). Four days after coculture, the proliferative T cells were analyzed by assessing the CFSE-diluted population.”
On Page 5 (paragraph 2.8.), lines 218-219:
“Statistical analysis was performed using GraphPad Prism version 7 (GraphPad Software Inc., CA, USA).” takes the place of this sentence “GraphPad Prism Software, version 7 (GraphPad Software Inc, CA, USA), was used for statistical analysis.”
On Page 7(paragraph 3.1.), lines 263-267:
“Adiponectin inhibited inflammatory cytokine and promoted M2 macrophage phenotype, including upregulation of IL-10 and arginase-1, in macrophages [29, 30]. Apo-A1, a well-known major structural protein of high-density lipoprotein (HDL), regulating cholesterol trafficking and affecting immune responses” takes the place of these sentences “Adiponectin reduced inflammatory cytokine production and enhanced M2 macrophage markers, such as IL-10 and arginase-1, in both human and mouse macrophages [29, 30]. Apo-A1, a well-known protein component of high-density lipoprotein (HDL), plays a role in cholesterol trafficking and immune regulation”
On Page 7 (paragraph 3.1.), lines 279-280:
“In summary, our findings demonstrate that Apo-A1 is markedly elevated in the UTUC microenvironment and promotes arginase-1 expression in neutrophils.” takes the place of this sentence “In summary, our findings indicate that Apo-A1 is significantly elevated in the UTUC microenvironment and plays a pivotal role in upregulating arginase-1 expression in neutrophils.”
On Page 16 (legend of Figure 5), lines 471-472:
“The Spearman method for rank correlation was applied to analyze the link between the two markers.” takes the place of this sentence “Spearman's rank-correlation test was used to determine the association between two markers.”
On Page 19 (Author Contributions), lines 614-615:
Correction “W.‑H. Lai: resources, data curation. C.-F. Hung: resources, data curation.”
On Page 19 (Institutional Review Board Statement), lines 623-624:
Correction “(No. 2020121)”

Reviewer 2 Report
Comments and Suggestions for Authors
Dear authors of “Immunosuppression of tumor-derived factors modulated neutrophils in upper tract urothelial carcinoma through upregulation of arginase-1 via ApoA1-STAT3 axis”,
Thank you for your contribution to this field. This is an interesting article aimed at precising the impact of neutrophils in upper tract urothelial carcinoma (UTUC).
The manuscript is well written. The research is conducted in a classical way, and the results obtained are quite convincing. Overall, the findings could be of interest for the UTUC field.
I have a small request before final acceptance: can you please also comment, in the discussion part, about the other up-regulated proteins in the UTUC biopsy tissues.
Especially about EMMPRIN (please write in capital letters line 230) and MMP-9, since those two proteins are involved in tumor metastasis and, as you mentioned, line 66, “60% of UTUC cases are invasive at diagnosis”. Also, EMMPRIN is well known for inducing neutrophil chemotaxis and migration.
Regards,
Author Response
Dear Editor and Reviewers:
We greatly thank you for the extensive assessment of our manuscript (cells-3601956), and for your invaluable comments. We have revised the original manuscript in accordance with reviewer’s comments, and corresponding changes are highlighted in red. Additionally, some sentences have been rewritten (highlighted in green) to reduce similarity with previously published papers, according to the editor’s suggestion. The Part I presents our responses (highlighted in blue) to reviewer’s comments (underline), while Part II outlines the revisions made in response to the editor’s suggestions or corrections of identified issues (highlighted in orange).
Part I:
Reviewer#2
Comments and Suggestions for Authors
- Dear authors of “Immunosuppression of tumor-derived factors modulated neutrophils in upper tract urothelial carcinoma through upregulation of arginase-1 via ApoA1-STAT3 axis”,
- Thank you for your contribution to this field. This is an interesting article aimed at precising the impact of neutrophils in upper tract urothelial carcinoma (UTUC).
- The manuscript is well written. The research is conducted in a classical way, and the results obtained are quite convincing. Overall, the findings could be of interest for the UTUC field.
- I have a small request before final acceptance: can you please also comment, in the discussion part, about the other up-regulated proteins in the UTUC biopsy tissues.
- Especially about EMMPRIN (please write in capital letters line 230) and MMP-9, since those two proteins are involved in tumor metastasis and, as you mentioned, line 66, “60% of UTUC cases are invasive at diagnosis”. Also, EMMPRIN is well known for inducing neutrophil chemotaxis and migration.
- Regards,
- Responses:
In response to the reviewer’s suggestion, we have discussed the factors involved in metastasis and added the following sentences to the revised manuscript:
On Page 18, lines 574-588
“Tumor associated neutrophils and macrophages could produce numerous protease, including MMPs, which contribute to extracellular matrix degradation, thereby supporting tumor development and metastasis [57]. EMMPRIN, a well-known inducer of MMPs, expresses in various tissue and regulates physiological and pathological processes [58]. Overexpression of EMMPRIN in tumor tissues has been associated with poor prognosis in bladder cancer [58] and is considered a potential prognostic factor in various other cancers [59]. EMMPRIN expression on the surface of neutrophils enhances MMPs production and chemotaxis, linking it to the pathogenesis of inflammatory diseases [60]. In many solid tumors, MMPs produced by stromal cells facilitate the release of soluble EMMPRIN from the plasma membrane of tumor cells, creating a positive feedback loop that further induces MMPs expression [58]. In this study, among the previously mentioned top three upregulated factors in UTUC, we focused on examining the role of Apo-A1 in immunosuppressive neutrophils, rather than investigating the other identified factors, including EMMPRIN and MMP-9. Therefore, the specific role of neutrophils in cancer metastasis and tumor development was not explored in our study.”
- Responses:
Additionally, other words have been added or modified in the revised manuscript:
On Page 2, line 78
“MMP-9 (matrix metalloproteinase-9)”
On Page 5, lines 231-232
“EMMPRIN (extracellular matrix metalloproteinase inducer)”
On Page 20 (Abbreviations)
“EMMPRIN extracellular matrix metalloproteinase inducer”
“MMP ematrix metalloproteinase”
Part II:
The revised words associated with editor’s suggestion or corrections of identified issues are described below:
On Page 3 (paragraph 2.1.), lines 121-122:
“The study enrolled participants from regular urological practices at the Chia-Yi Christian Hospital.” takes the place of this sentence “Study participants were recruited from customary urological practices at the Chia-Yi Christian Hospital.”
On Page 3 (paragraph 2.1.), lines 124-129:
“Table 1 presents hematological data, including proportions of peripheral neutrophil and lymphocyte, as well as the characteristics of the study participants. Ethical approval was obtained from the Chia-Yi Christian Hospital Ethics Committee (No. 2020121), and the study was conducted in accordance with the ethical principles outlined in the Declaration of Helsinki for medical research involving human subjects.” takes the place of these sentences “The hematological profiles provided the data on the proportion of neutrophils and lymphocytes in peripheral bloods of study participants (Table 1). This study was approved by the Ethics Committee of Chia-Yi Christian Hospital (No. 2020061) in Taiwan and followed the Declaration of Helsinki ethical principles for medical research involving human subjects.”
On Page 3 (paragraph 2.2.), lines 133-134:
“surgical isolated tumor tissue was washed with 1× PBS for elimination of residual mucus and blood” takes the place of this sentence “surgical isolated tumor tissue was washed with 1× PBS to remove blood and mucus”
On Page 3 (paragraph 2.2.), lines 135-138:
“RPMI1640 medium supplemented with penicillin (100 units/mL, HIMEDIA, Mumbai, India), streptomycin (100 μg/mL, HIMEDIA, Mumbai, India) and 10% fetal bovine serum (FBS, Gibco, NY, USA). The culture was incubated for 24 hours using 4 mL of medium per gram of tumor,” takes the place of this sentence “RPMI1640 medium (HIMEDIA, Mumbai, India) containing 10% fetal bovine serum (FBS, Gibco, NY, USA), 100 unit/mL penicillin (HIMEDIA, Mumbai, India), and 100 μg/mL streptomycin (HIMEDIA, Mumbai, India) for 24 hr (4 mL medium/ g tumor),”
On Page 3 (paragraph 2.2.), lines 141-144:
“After resuspending the cells in culture medium, Ficoll-Paque gradient centrifugation was performed.” takes the place of these sentences “The cells were resuspended with culture medium and subsequently subjected to Ficoll-Paque gradient centrifugation.”
On Page 4 (paragraph 2.4.), lines 157-158:
“RPMI1640 medium supplemented with 100 unit/mL penicillin, 100 μg/mL streptomycin and 10% FBS.” takes the place of this sentence “RPMI1640 medium containing 10% FBS, 100 unit/mL penicillin, and 100 μg/mL streptomycin.”
On Page 4 (paragraph 2.4.), lines 173-178:
“Neutrophils were cocultured with carboxyfluorescein succinimidyl ester (CFSE, ThermoFisher, Oregon, USA)-labeled CD4⁺ or CD8⁺ T cells (neutrophil: T=1:1), in 96 well plates. The coculture wells were incubated with anti-CD3/CD28 Abs (DynabeadsTM human T activator CD3/CD28, ThermoFisher, MA, USA) for T cell stimulation. Four days after coculture, CFSE-diluted signal was assessed for determination of proliferative T cell population.” takes the place of these sentences “Neutrophils were cocultured with carboxyfluorescein succinimidyl ester (CFSE, ThermoFisher, Oregon, USA)-labeled CD4⁺ or CD8⁺ T cells, at a ratio of 1:1, in 96 well plates, upon anti-CD3/CD28 stimulation (DynabeadsTM human T activator CD3/CD28, ThermoFisher, MA, USA). Four days after coculture, the proliferative T cells were analyzed by assessing the CFSE-diluted population.”
On Page 5 (paragraph 2.8.), lines 218-219:
“Statistical analysis was performed using GraphPad Prism version 7 (GraphPad Software Inc., CA, USA).” takes the place of this sentence “GraphPad Prism Software, version 7 (GraphPad Software Inc, CA, USA), was used for statistical analysis.”
On Page 7(paragraph 3.1.), lines 263-267:
“Adiponectin inhibited inflammatory cytokine and promoted M2 macrophage phenotype, including upregulation of IL-10 and arginase-1, in macrophages [29, 30]. Apo-A1, a well-known major structural protein of high-density lipoprotein (HDL), regulating cholesterol trafficking and affecting immune responses” takes the place of these sentences “Adiponectin reduced inflammatory cytokine production and enhanced M2 macrophage markers, such as IL-10 and arginase-1, in both human and mouse macrophages [29, 30]. Apo-A1, a well-known protein component of high-density lipoprotein (HDL), plays a role in cholesterol trafficking and immune regulation”
On Page 7 (paragraph 3.1.), lines 279-280:
“In summary, our findings demonstrate that Apo-A1 is markedly elevated in the UTUC microenvironment and promotes arginase-1 expression in neutrophils.” takes the place of this sentence “In summary, our findings indicate that Apo-A1 is significantly elevated in the UTUC microenvironment and plays a pivotal role in upregulating arginase-1 expression in neutrophils.”
On Page 16 (legend of Figure 5), lines 471-472:
“The Spearman method for rank correlation was applied to analyze the link between the two markers.” takes the place of this sentence “Spearman's rank-correlation test was used to determine the association between two markers.”
On Page 19 (Author Contributions), lines 614-615:
Correction “W.‑H. Lai: resources, data curation. C.-F. Hung: resources, data curation.”
On Page 19 (Institutional Review Board Statement), lines 623-624:
Correction “(No. 2020121)”

Reviewer 3 Report
Comments and Suggestions for Authors
The main question addresses the investigation of mechanisms inducing changes in immune response in the presence of upper transitional urothelial carcinoma, what may lead to cancer progression and be helpful with assessment of response to patient immunotherapy.
The article shows original investigations of in-vitro expression of various proteins in neutrophils, and in-vivo neutrophils activation (in the presence or absence of urothelial carcinoma) effect on lymphocyte infiltration in tumour (CD8 cytotoxic, and CD4 helper).
Compared with other published material, the article shows the changes in neutrophils (arginase-1) and lymphocytic reaction particularly in cases with the upper transitional urothelial carcinoma (with or not of STAT6 expression).
The authors should consider performing animal model experiments with the addition of immunotherapy, and examine changes to the various cell types counts and activaction.
However, the figures need corrections (space-planning; the graphs from GraphPad are copied into lines of the text and named subsequently, e.g. Figure 2 A, B, ...G - moreover, that figure contains even 3 pages).
The conclusions are consistent with the evidence and arguments presented, and do they address the main thesis.
The references are adequate.
Author Response
Dear Editor and Reviewers:
We greatly thank you for the extensive assessment of our manuscript (cells-3601956), and for your invaluable comments. We have revised the original manuscript in accordance with reviewer’s comments, and corresponding changes are highlighted in red. Additionally, some sentences have been rewritten (highlighted in green) to reduce similarity with previously published papers, according to the editor’s suggestion. The Part I presents our responses (highlighted in blue) to reviewer’s comments (underline), while Part II outlines the revisions made in response to the editor’s suggestions or corrections of identified issues (highlighted in orange).
Part I:
Reviewer#3
Comments and Suggestions for Authors
The main question addresses the investigation of mechanisms inducing changes in immune response in the presence of upper transitional urothelial carcinoma, what may lead to cancer progression and be helpful with assessment of response to patient immunotherapy.
The article shows original investigations of in-vitro expression of various proteins in neutrophils, and in-vivo neutrophils activation (in the presence or absence of urothelial carcinoma) effect on lymphocyte infiltration in tumour (CD8 cytotoxic, and CD4 helper).
Compared with other published material, the article shows the changes in neutrophils (arginase-1) and lymphocytic reaction particularly in cases with the upper transitional urothelial carcinoma (with or not of STAT6 expression).
The authors should consider performing animal model experiments with the addition of immunotherapy, and examine changes to the various cell types counts and activaction.
- Responses:
The advancement of UTUC research is hindered not only by its rarity but also by the lack of disease-specific models [1,2]. The xenograft model in nude mice, established using a human UTUC cell line, has been used to evaluate the synergistic effects of selective drugs in combination with paclitaxel [3]. Additionally, patient-derived UTUC tumor cell lineages have been applied in the development of personalized medicine through the use of an immunocompromised mouse model [1]. However, these animal models cannot mimic the modulation of T cells within the tumor microenvironment of UTUC. A recent study proposed a novel mouse model, established with female BALB/c mice, which develops UTUC rather than bladder cancer after carcinogen BBN induction [4]. However, this tumor displayed basal-type markers and was enriched with increased T cells and macrophages [4], which is not consistent with the predominant luminal-type in UTUC [1,2]. As a result, the investigation of immunomodulation in UTUC tumors, especially the role of neutrophils, remains limited.
However, the figures need corrections (space-planning; the graphs from GraphPad are copied into lines of the text and named subsequently, e.g. Figure 2 A, B, ...G - moreover, that figure contains even 3 pages).
The conclusions are consistent with the evidence and arguments presented, and do they address the main thesis.
The references are adequate.
- Responses:
In response to the reviewer’s suggestion, we have rearranged most of figures to separate the graphs and assign clear, corresponding labels.
References
- Kim, K.; Hu, W.; Audenet, F.; Almassi, N.; Hanrahan, A.J.; Murray, K.; Bagrodia, A.; Wong, N.; Clinton, T.N.; Dason, S.; et al. Modeling biological and genetic diversity in upper tract urothelial carcinoma with patient derived xenografts. Nat. Commun. 2020, 11, 1975, doi:10.1038/s41467-020-15885-7.
- Li, Z.; Xu, H.; Gong, Y.; Chen, W.; Zhan, Y.; Yu, L.; Sun, Y.; Li, A.; He, S.; Guan, B.; et al. Patient-Derived Upper Tract Urothelial Carcinoma Organoids as a Platform for Drug Screening. Adv Sci (Weinh) 2022, 9, e2103999, doi:10.1002/advs.202103999.
- Hsu, F.S.; Wu, J.T.; Lin, J.Y.; Yang, S.P.; Kuo, K.L.; Lin, W.C.; Shi, C.S.; Chow, P.M.; Liao, S.M.; Pan, C.I.; et al. Histone Deacetylase Inhibitor, Trichostatin A, Synergistically Enhances Paclitaxel-Induced Cytotoxicity in Urothelial Carcinoma Cells by Suppressing the ERK Pathway. Int. J. Mol. Sci. 2019, 20, doi:10.3390/ijms20051162.
- Yamamoto, A.; Kawashima, A.; Uemura, T.; Nakano, K.; Matsushita, M.; Ishizuya, Y.; Jingushi, K.; Hase, H.; Katayama, K.; Yamaguchi, R.; et al. A novel mouse model of upper tract urothelial carcinoma highlights the impact of dietary intervention on gut microbiota and carcinogenesis prevention despite carcinogen exposure. Int J Cancer 2025, 156, 1439-1456, doi:10.1002/ijc.35295.
Part II:
The revised words associated with editor’s suggestion or corrections of identified issues are described below:
On Page 3 (paragraph 2.1.), lines 121-122:
“The study enrolled participants from regular urological practices at the Chia-Yi Christian Hospital.” takes the place of this sentence “Study participants were recruited from customary urological practices at the Chia-Yi Christian Hospital.”
On Page 3 (paragraph 2.1.), lines 124-129:
“Table 1 presents hematological data, including proportions of peripheral neutrophil and lymphocyte, as well as the characteristics of the study participants. Ethical approval was obtained from the Chia-Yi Christian Hospital Ethics Committee (No. 2020121), and the study was conducted in accordance with the ethical principles outlined in the Declaration of Helsinki for medical research involving human subjects.” takes the place of these sentences “The hematological profiles provided the data on the proportion of neutrophils and lymphocytes in peripheral bloods of study participants (Table 1). This study was approved by the Ethics Committee of Chia-Yi Christian Hospital (No. 2020061) in Taiwan and followed the Declaration of Helsinki ethical principles for medical research involving human subjects.”
On Page 3 (paragraph 2.2.), lines 133-134:
“surgical isolated tumor tissue was washed with 1× PBS for elimination of residual mucus and blood” takes the place of this sentence “surgical isolated tumor tissue was washed with 1× PBS to remove blood and mucus”
On Page 3 (paragraph 2.2.), lines 135-138:
“RPMI1640 medium supplemented with penicillin (100 units/mL, HIMEDIA, Mumbai, India), streptomycin (100 μg/mL, HIMEDIA, Mumbai, India) and 10% fetal bovine serum (FBS, Gibco, NY, USA). The culture was incubated for 24 hours using 4 mL of medium per gram of tumor,” takes the place of this sentence “RPMI1640 medium (HIMEDIA, Mumbai, India) containing 10% fetal bovine serum (FBS, Gibco, NY, USA), 100 unit/mL penicillin (HIMEDIA, Mumbai, India), and 100 μg/mL streptomycin (HIMEDIA, Mumbai, India) for 24 hr (4 mL medium/ g tumor),”
On Page 3 (paragraph 2.2.), lines 141-144:
“After resuspending the cells in culture medium, Ficoll-Paque gradient centrifugation was performed.” takes the place of these sentences “The cells were resuspended with culture medium and subsequently subjected to Ficoll-Paque gradient centrifugation.”
On Page 4 (paragraph 2.4.), lines 157-158:
“RPMI1640 medium supplemented with 100 unit/mL penicillin, 100 μg/mL streptomycin and 10% FBS.” takes the place of this sentence “RPMI1640 medium containing 10% FBS, 100 unit/mL penicillin, and 100 μg/mL streptomycin.”
On Page 4 (paragraph 2.4.), lines 173-178:
“Neutrophils were cocultured with carboxyfluorescein succinimidyl ester (CFSE, ThermoFisher, Oregon, USA)-labeled CD4⁺ or CD8⁺ T cells (neutrophil: T=1:1), in 96 well plates. The coculture wells were incubated with anti-CD3/CD28 Abs (DynabeadsTM human T activator CD3/CD28, ThermoFisher, MA, USA) for T cell stimulation. Four days after coculture, CFSE-diluted signal was assessed for determination of proliferative T cell population.” takes the place of these sentences “Neutrophils were cocultured with carboxyfluorescein succinimidyl ester (CFSE, ThermoFisher, Oregon, USA)-labeled CD4⁺ or CD8⁺ T cells, at a ratio of 1:1, in 96 well plates, upon anti-CD3/CD28 stimulation (DynabeadsTM human T activator CD3/CD28, ThermoFisher, MA, USA). Four days after coculture, the proliferative T cells were analyzed by assessing the CFSE-diluted population.”
On Page 5 (paragraph 2.8.), lines 218-219:
“Statistical analysis was performed using GraphPad Prism version 7 (GraphPad Software Inc., CA, USA).” takes the place of this sentence “GraphPad Prism Software, version 7 (GraphPad Software Inc, CA, USA), was used for statistical analysis.”
On Page 7(paragraph 3.1.), lines 263-267:
“Adiponectin inhibited inflammatory cytokine and promoted M2 macrophage phenotype, including upregulation of IL-10 and arginase-1, in macrophages [29, 30]. Apo-A1, a well-known major structural protein of high-density lipoprotein (HDL), regulating cholesterol trafficking and affecting immune responses” takes the place of these sentences “Adiponectin reduced inflammatory cytokine production and enhanced M2 macrophage markers, such as IL-10 and arginase-1, in both human and mouse macrophages [29, 30]. Apo-A1, a well-known protein component of high-density lipoprotein (HDL), plays a role in cholesterol trafficking and immune regulation”
On Page 7 (paragraph 3.1.), lines 279-280:
“In summary, our findings demonstrate that Apo-A1 is markedly elevated in the UTUC microenvironment and promotes arginase-1 expression in neutrophils.” takes the place of this sentence “In summary, our findings indicate that Apo-A1 is significantly elevated in the UTUC microenvironment and plays a pivotal role in upregulating arginase-1 expression in neutrophils.”
On Page 16 (legend of Figure 5), lines 471-472:
“The Spearman method for rank correlation was applied to analyze the link between the two markers.” takes the place of this sentence “Spearman's rank-correlation test was used to determine the association between two markers.”
On Page 19 (Author Contributions), lines 614-615:
Correction “W.‑H. Lai: resources, data curation. C.-F. Hung: resources, data curation.”
On Page 19 (Institutional Review Board Statement), lines 623-624:
Correction “(No. 2020121)”
